# Characterization of a Sandwich PLGA-Gallic Acid-PLGA Coating on Mg Alloy ZK60 for Bioresorbable Coronary Artery Stents

**DOI:** 10.3390/ma13235538

**Published:** 2020-12-04

**Authors:** Li-Han Lin, Hung-Pang Lee, Ming-Long Yeh

**Affiliations:** 1Department of Biomedical Engineering, National Cheng Kung University, Tainan 701, Taiwan; d09522010@ntu.edu.tw; 2Biomedical Engineering, Dwight Look College of Engineering, Texas A&M University, College Station, TX 77843, USA; qer6322129@tamu.edu; 3Medical Device Innovation Center, National Cheng Kung University, Tainan 701, Taiwan

**Keywords:** magnesium alloy, cardiovascular stents, callic acid, dip coating, endothelialization, anticorrosion

## Abstract

Absorbable magnesium stents have become alternatives for treating restenosis owing to their better mechanical properties than those of bioabsorbable polymer stents. However, without modification, magnesium alloys cannot provide the proper degradation rate required to match the vascular reform speed. Gallic acid is a phenolic acid with attractive biological functions, including anti-inflammation, promotion of endothelial cell proliferation, and inhibition of smooth muscle cell growth. Thus, in the present work, a small-molecule eluting coating is designed using a sandwich-like configuration with a gallic acid layer enclosed between poly (d,l-lactide-co-glycolide) layers. This coating was deposited on ZK60 substrate, a magnesium alloy that is used to fabricate bioresorbable coronary artery stents. Electrochemical analysis showed that the corrosion rate of the specimen was ~2000 times lower than that of the bare counterpart. The released gallic acid molecules from sandwich coating inhibit oxidation by capturing free radicals, selectively promote the proliferation of endothelial cells, and inhibit smooth muscle cell growth. In a cell migration assay, sandwich coating delayed wound closure in smooth muscle cells. The sandwich coating not only improved the corrosion resistance but also promoted endothelialization, and it thus has great potential for the development of functional vascular stents that prevent late-stent restenosis.

## 1. Introduction

According to reports from the Global Health Organization in 2016, cardiovascular diseases (CVDs), the prevalence of which increased by 15% in the past decade, are the leading cause of death (44%) among non-communicable diseases [1]. In clinical settings, a percutaneous coronary intervention (PCI) combined with balloon angioplasty and stent implantation is the gold standard for treating stenotic arteries. Currently, dual antiplatelet therapy is suggested for at least 12 months to alleviate mural thrombosis caused by PCI [2]. The traditional materials used in cardiovascular stents are 316 L stainless steels, cobalt-chromium alloys, and nickel-titanium alloys [3]. However, the permanent installation of devices inside the human body causes a chronic inflammatory local reaction and long-term endothelial dysfunction [4]. Biodegradable magnesium (Mg) and its alloys are ideal candidates for stent platforms because they have less neointimal formation and provide long-term inhibition of the growth of smooth muscle cells (SMCs), unlike stainless steel [5,6]. The Mg alloys also have twice the tensile strength of unmodified biodegradable poly-l-lactide (PLLA), which leads to flexible stenting deployment and better radial force [7,8,9].

To properly use a magnesium alloy as a biodegradable stent platform for physiological applications, the corrosion rate must be controlled. To date, the surfaces of degradable magnesium alloys have been modified in several ways, such as alkaline heat treatment (AHT), the sol-gel process, chemical conversion, and micro-arc oxidation (MAO) [10,11]. Although ceramic MAO processing may provide a consolidated surface [12,13,14] for clinical applications, a prior study showed that this process did not improve corrosion resistance after three months [15]. In contrast, a sol-gel coating can greatly improve the corrosion resistance with chemical modifications [16]. Polymetric processes create a physical barrier [17] that enhances the corrosion resistance of Mg-stents, and the barrier also serves as an absorbable drug-loading system [18,19]. A polymer coating of a specific thickness can prevent complete degradation of Mg alloys for periods of 1 to 12 months [20].

Phenolic molecules, such as gallic acids (GA), exert and induce specific levels of selective viability in human endothelial cells (ECs) and SMCs [21,22,23,24]. Furthermore, these extracted molecules have anti-oxidation characteristics that inhibit inflammation. Since cellular antioxidants prevent the formation of oxidized low-density lipoprotein (LDL) and antiplatelet activation, this is a remarkable method by which to treat atherosclerosis [25]. Although several articles have indicated that the phenolic conversion coating on a Mg-alloy is non-toxic and anti-corrosive, chemical conversion may compromise stent integrity through the phenolic-Mg conversion process [26,27,28,29].

Mg-Zn alloys are well-known for their good corrosion resistance and physiological safety [30]. The ZK60 Mg-6Zn-0.5Zr alloy has been used to make vascular stents due to its better biocompatibility than those of other Mg alloys [31]. In addition, ZK60 has a high tensile strength (315 MPa) that prolongs the radial force of Mg stents. Nevertheless, the rapid degradation rate of bare ZK60 represents a shortcoming of the alloy for its use to fabricate bioresorbable coronary artery stents [15]. In this paper, a coating comprising of GA sandwiched between two layers of Poly (d,l-lactide-co-glycolide) (PLGA), a novel sandwich coating, on Mg alloy ZK60 concept was proposed. It can serve not only as a protective barrier that increases corrosion resistance but also as a reservoir for controlling GA release. The surface properties of coated-Mg stent and corrosion behavior of a Mg alloy were determined. Furthermore, cell viability and cell migration tests were performed to investigate the cell regulation of released GA in endothelialization as well as the effects on the growth of smooth muscle cells.

## 2. Materials and Methods

### 2.1. Materials and Specimen Preparation

In this study, extruded commercial ZK60 alloy bars (Zn 5.48 wt.%, Zr 0.42 wt.% and balance Mg) comprised the starting material for the substrate [32]. PLGA in molar ratio of 85/15 and GA with purity above 97.5% were purchased from Sigma-Aldrich (St. Louis, MO, USA; Figure 1a). ZK60 disks cut from the bars were reduced to 12 mm in diameter and 5 mm in thickness (Figure 1c). These specimens were ground with silicone carbide sandpaper (150–5000 mesh) and polished. Before being dried in a stream of air, all specimens were rinsed ultrasonically with acetone, ethanol, and distilled water for 5 min, respectively. Mg-stent was machined by an INTAI Technology and CHONG HUAI laser (Taichung, Taiwan) for further evaluation.

### 2.2. Sandwich Coating Films Preparation

Mirror-polished ZK60 specimens were soaked in an alkaline solution composed of 20 wt.% NaOH. The solution was stirred at 240 rpm and then heated for 90 min at 60 °C to equilibrate the specimens. After 90 min, the specimens were drawn out of the alkaline solution and rinsed with deionized water. The rinsed specimens were incubated at 80 °C for 30 min until they were dried. Subsequently, heat treatment at 120 °C for another 30 min was conducted to stabilize the oxide film. To fabricate the sol-gel PLGA coating layer, PLGA was dissolved at a concentration of 4 wt.% in 10 mL of chloroform. The PLGA film thickness depended on the drawing speed of 3 mm/s, and the film was dried in a stream of air to form a uniform dip-coating surface. Subsequently, different coating layers were prepared with or without GA solution. The samples were dipped into GA solution at a concentration of 1 wt.% and 10 mL acetone. Finally, the samples were ultrasonically rinsed with ethanol and air-dried. The coating steps were conducted on the Mg-stent for surface morphology observation.

### 2.3. Characterization of the Surface, the Cross-Section Structure, and the Elements Content

The crystallinity of ZK60 was analyzed using thin film X-ray diffraction (TF-XRD Bruker D8 Discover, Brucker, Karlsruhe, Germany) with Cu-Kα radiation. Diffraction patterns were acquired at 2θ values of 20–80°. The surface morphology and element distributions of the coating films were studied by scanning electron microscopy (SEM JSM-6700F, JEOL, Tokyo, Japan) under 10 kV acceleration voltage and energy dispersive spectrometry (EDS JSM-6700F, JEOL, Tokyo, Japan) under 0.2 keV acquisition energy, respectively. The structures of the films were recorded with a Fourier transform-infrared (FT-IR) spectrophotometer (FTIR-4600, Jasco, Tokyo, Japan) at a transmitter ratio (T%) and infra spectra resolution of 4 cm^−1^. The spectra were collected in the range of 600–4000 cm^−1^.

### 2.4. Electrochemical Corrosion and Hydrogen Evolution Tests

The electrochemical and hydrogen evolution tests were performed in revised simulated body fluid (r-SBF) solution (per liter, 0.072 g of NaSO_4_, 0.182 g of K_2_HPO_4_, 0.225 g of KCl, 0.310 g of MgCl_2_ 6H_2_O, 0.736 g of NaHCO_3_, 0.923 g of CaCl_2_, 2.036 g Na_2_CO_3_, 5.403 g of NaCl, and 11.928 g of 4-(2-hydroxyethyl)-1-piperazineethanesulfonic avid (HEPES) dissolved in deionized water), respectively. In the electrochemical corrosion tests, a PARSTAT 2273 electrochemistry workstation (PARSTAT 2273, AMETEK, Berwyn, PA, USA) was operated at a scanning rate of 1 mV s^−1^ at −2.0 V to 1.0 V with a step height of 2.5 mV [12]. A reference saturated calomel electrode (SCE KCl) combined with a conventional three-electrode electrode cell and a platinum plate was used for the electrochemical analysis. The area of the working electrode exposed to the electrolyte was controlled by a polylactic acid (PLA) holder to within 1 cm^2^. Each sample was tested for once and immersed in r-SBF solution at least three hours [12]. In hydrogen evolution test, all samples (n = 3) were freshly prepared and then placed in the r-SBF solution for 48 h at a pH of 7.4 and a temperature of 37 °C. The equipment then recorded the total volume of the hydrogen released from the magnesium alloys [32].

### 2.5. Cytocompatibility Evaluation

The cytocompatibility tests examined cytotoxicity, cell proliferation, and migration. A human umbilical vein cell line (EA. hy926) and a rat aortic smooth muscle cell line (RASMC) were provided by Wen-Tai Chiu (National Cheng Kung University, Tainan, Taiwan). The growth behaviors of the EA. hy926 and RASMC represented the cardiovascular ECs and SMCs, respectively. Before cell culturing, the sterile samples were immersed in free fetal bovine serum Dulbecco’s Modified Eagle Medium (FBS DMEM) (approximately 1.25 cm^2^/mL in DMEM) for 24-h extraction. The resulting filtered (0.2 µm filter) medium was then diluted to a 15 mL volume. The cells were seeded separately at a density of 4000 cells/well in 96-well plates. The 90% DMEM with 10% dimethyl sulfoxide (DMSO) was used as the positive control, and 90% DMEM with 10% FBS was used as the negative control. The culture medium was replaced with 90% extracted medium with 10% FBS overnight. Before collection of the optical density (OD) values at 450 nm, cell counting kit-8 (CCK8) solution was added into each well and the mixture was incubated for 2 h. Briefly, the cell proliferation was recorded on days 1, 4, and 7. Three replications were conducted and then the results were calculated using Equation (1) below.
Relative growth rate % = [(OD test − OD positive)/(OD negative − OD positive)] × 100%(1)

### 2.6. Hemolysis Tests

The hemolysis tests were performed according to the ISO 10993-4 standard for biomaterials [33]. Sodium citrate (4 wt.%) to the fresh blood samples in the ratio of 1:9 was taken 30 min before the tests. All specimens were immersed in centrifuge tubes containing 10 mL of normal saline and incubated for 30 min at 37 °C. Deionized water was used as the positive control, and normal saline was used as the negative control. After 30 min of incubating, 0.2 mL of the diluted blood, prepared with normal saline at a volume ratio of 4:5, was added into the centrifuge tubes, and all the tubes were incubated for 60 min at 37 °C. After 60 min, the tubes were centrifuged for 5 min at 2500 rpm, and the supernatant was collected and carefully transferred to a 96-well plate for spectroscopic measurement. The hemolysis data, read by ELISA, were calculated using Equation (2) below and were based on the average of three replications:Hemolysis % = [(OD test − OD negative)/(OD positive − OD negative)] × 100%(2)

### 2.7. Free Radical Activity Tests

The free radical activity tests, measured diphenyl-2-picrylhydrazyl (DPPH), were performed according to application on bleached teeth and several antioxidant tests on magnesium alloys [34]. The DPPH powder was diluted in 0.2 mmol/mL DPPH solution with 70% ethanol. All specimens were immersed in 2 mL of DPPH solution and incubated at 37 °C for 1 h in the dark. Similarly, DPPH solution was used as the control group. Then the absorbance of DPPH data, read by microplate reader, were calculated using Equation (3) below based on the average of three replications:Inhibition % = [(OD control − OD test)/OD control] × 100%(3)

### 2.8. Statistical Analysis

In this study, all quantitative results are expressed as standard deviation (SD) unless indicated. Each assay was performed in at least three replicated tests, as described above. Measured experimental results from GraphPad Prism software (Prism 9.0, GraphPad, San Diego, CA, USA) were analyzed by a non-parametric test (Kruskal-Wallis Test) combined with Uncorrected Dunn’s multiple comparisons test (each comparison stands alone), and a *p*-value < 0.05 was considered significant.

## 3. Results

### 3.1. Modification of the ZK60 Surface

An as-extruded ZK60 disk scanned by XRD indicated three peaks at 32.2°, 34.5°, and 36.8°, corresponding to Mg in Appendix A (Figure A1). As-extruded ZK60 cut disks were modified with AHT to develop a layer of magnesium hydroxide as Equation (4) below. The Mg(OH)_2_ layer was then treated at high temperature to form a rough MgO layer as Equation (5) following by the PLGA dip coating. These chemical reactions produced a less-active ZK60 surface and a compact void-free oxide for the dip-coating process. The void-free structure prevented inner bulk erosion on the polymer layer and external pitting corrosion on the highly active magnesium alloy. In comparison to the bare ZK60, the MgO coating had higher corrosion resistance to delay magnesium ions from bursting out in Appendix A (Figure A2). The sandwich coating steps were schematized and are depicted in order in Figure 1b. As illustrated in that figure, the bare ZK60 substrate was first treated with AHT and then coated with sandwich coating layers. Since hydrophilic GA was unable to dissolve in the oil phase, the hydrophobic PLGA was able to remain compact in the sandwich layer coating process.

The FT-IR analysis confirmed the encapsulation of GA in the polymer sandwich layers (Figure 1c). The wavenumbers of the functional group in GA were sourced from a published FT-IR article [35]. The presence of the benzene signal at 1482 cm^−1^ allowed a decisive characterization of the phenolic GA compound even though PLGA and GA shared several similar functional groups. The GA group had benzene vibration peaks at 1482 cm^−1^, and PLGA did not, while the interference of C=O (1750 cm^−1^) and C-O (1250–1100 cm^−1^) vibration peaks caused a broad benzene peak of the sandwich coating from 1600 cm^−1^ to 1300 cm^−1^. Also, the O-H peaks, representing tri-hydroxyl groups at 3683 cm^−1^, provided substantial evidence that the sandwich coating immobilized the GA in the sandwich structure.
(4)$$Mg^{2+}\left(aq\right)+2OH^{-}\left(aq\right)⇌Mg{\left(OH\right)}_{2}\left(s\right)$$
(5)$$Mg{\left(OH\right)}_{2}\left(s\right)\overset{\Delta }{\to }MgO\left(s\right)+H_{2}O\left(g\right)↑$$

### 3.2. Effects of PLGA Dip-Coating and Phenolic Layer on the Coating Morphology

The SEM images of the sandwich coating showed that the AHT-modified ZK60 surface became more uniform after a series of dip-coating processes (Figure 2). Initially, the morphology of the MgO after AHT was rough and coarse. After the PLGA dip-coating, a non-homogeneous microstructure formed on the MgO surface. After the second PLGA dip-coating, the top PLGA film layer smoothed the rough GA layer in the form of a sandwiched-layer structure. In addition, the SEM images indicated that PLGA and sandwich coatings had film thicknesses of 1.1 ± 0.4 µm and 2.1 ± 0.3 µm, respectively, in the cross-section view. The thickness of the coating’s cross-sections was proportionate with the number of PLGA layers.

EDS detection confirmed that the sandwich layers covered the bare ZK60, as shown in the elemental distribution in Figure 3a. In the MgO layer, magnesium and oxygen accounted for 36.9% and 63.1%, respectively, while in the final sandwich coating film, Mg and oxygen (O) accounted for 7.2% and 23.4%, respectively, with the balance being carbon (C). The decreases in the contents of Mg and O with the layers verified that the PLGA coating protected the MgO surface. Furthermore, the rising C contents of GA and PLGA indicated bare Mg substrate was covered by the coating. Next, an EDS line scan was applied to analyze the elemental changes in the sandwich coating (Figure 3b,c). Based on the changes in the ratios of the element weights, the highest content of Mg was in the ZK60 region. Due to the rich oxygen contents of MgO, GA, and PLGA, the increasing O and declining Mg contents indicated the MgO layer and sandwich coating. Furthermore, the rising carbon signal originated from the mounting epoxy resin. The film thickness observed in the SEM images was consistent with the EDS analysis measured from the first Mg–O crossing point to the second Mg–C crossing point in the PLGA and sandwich coating films.

To evaluate the dip-coating effect on the ZK60 stent platform, the coating steps mentioned above were repeated on the ZK60 stent prototype. Although the ZK60 stent developed irregular corrosive struts without the coating, the coated Mg-stent retained its stent integrity, and a uniform polymer coating formed on its surface (Figure 4).

### 3.3. Anti-Corrosion Behavior

The hydrogen evolution was recorded for 48 h. The results indicated the highest H_2_ volume for the untreated ZK60, and the lowest, for sandwich coating (Figure 5a). The electrochemical analysis showed results comparable to those for hydrogen release. The lowest corrosion density of the sandwich coating, acquired from the potentiodynamic polarization curves, suggested that the sandwich coating improved the corrosion resistance (Figure 5b). The unmodified ZK60 had the lowest corrosion potential, −1.6 V, and the highest corrosion current density, 20.51 µA/cm^2^, which corresponded to the H_2_ release, indicating that the bare surfaces did suffer a severe corrosion attack in the physiological environment (Table 1). Furthermore, the PLGA had a higher corrosion potential, −0.4 V, and a lower corrosion current density, 1.79 µA/cm^2^, and the sandwich coating also exhibited a slightly increased corrosion potential of −0.2 V and a decreased corrosion current density of 0.01 µA/cm^2^ than those of the bare ZK60. These findings suggested a significant difference between PLGA and sandwich coating in terms of electrochemical performance. This sandwich structure provided a better corrosion resistance.

### 3.4. Effect of Phenolic Molecules on ECs and SMCs in TERMS of Cell Viability and Hemolysis

The sandwich coating layers promoted EC proliferation but inhibited the growth of SMCs (Figure 6). Changes in GA gradient resulted in varying EC vitality, demonstrating the relationship between EC tolerance and GA toxicity in Appendix A (Figure A3). The outcome demonstrated that the GA 1 wt.% group had better viability than did the other two groups, while the highly-concentrated GA suggested that direct over-exposure could cause apoptosis due to the high antioxidant activity. After comparison to different groups, including bare (ZK60), PLGA and sandwich coating, at 1, 4, and 7 days, sandwich coating showed no toxicity to ECs (Figure 6c).

Based on the cell viability of the ECs and SMCs, there were two notable points: the non-toxic growth rate of sandwich coating towards ECs, and the selective suppression of SMCs (Figure 6a,b). In comparison to bare ZK60, the PLGA and sandwich coatings exhibited significant viability in ECs. The SMCs exhibited proliferation in the PLGA group, while sandwich coating only had 90% viability after four days. Even though the proliferation of the ECs declined on Day 4, the decline could have been due to the high content of GA released in a static environment. The efficiency of GA, while not significant in the SMCs, was moderate in strength. These results indicated that the ECs had a robust proliferation growth rate of 150% compared to the SMC group, and sandwich coating inhibited SMC over-growth. None of the groups exhibited toxicity in the comparison tests; thus, the four-day test was sufficient to observe the growth trend, while a seven-day test would have soon reached 100% coverage and lost the trend in our relative growth study.

Sandwich coating presented a hemolysis ratio below 5%, corresponding to the clinical bio-safety standard (Figure 6d). The bare ZK60 had a high hemolysis ratio of more than 43%, while both sandwich coating and PLGA showed lower hemolysis rates of 2.1% and 3.8%, respectively. Overall, the sandwich demonstrated a promising ability to regulate ECs and SMCs with excellent hemocompatibility.

### 3.5. Anti-Oxidation

The linear regression retrieved from a gradient GA content test defined a standard reference of GA released weight and concentration (Figure 7a). The sandwich coating film triggered GA weight release at 1, 2, 3, and 6 h, with a reduced speed after three hours (Figure 7b). The free radical scavenging analysis showed that the bare, PLGA, and sandwich coatings could eliminate oxidant stress with antioxidant capacities of approximately 28%, 36%, and 63%, respectively. These data verified that the GA released from the sandwich coating film promoted scavenging of free radicals and protected the vascular tissue due to its significant anti-oxidative ability, suggesting that the sandwich coating is a promising material for vascular stents (Figure 7c).

### 3.6. Cell Migration

The sandwiched-layer structure ensured that the GA had an anti-proliferative effect on the SMCs and a slight influence on EC migration (Figure 8). Re-endothelialization at the lesion site is a particularly important standard after PCI and requires healthy EC proliferation, migration, and spreading. Additionally, ideal regulation of SMCs and ECs could prevent penetration from SMCs as well as late-stent restenosis. As the previous viability results showed, GA has a specific sensitivity to SMCs and ECs at GA−1wt.% (~4 µg/mL) in the sandwich-structured film. The results indicated a robust EC migration without a significant difference in migration length among all groups (Figure 8a). In contrast, the SMCs had a larger migration distance in the PLGA group than in the other groups, while the sandwich and control groups were alike (Figure 8b).

An overall analysis of the ECs and SMCs after 48 h showed 81% and 42% migration from sides to the middle wound closure, respectively, in the control group. In the PLGA group, both the ECs and SMCs exhibited migration of 90% and 55%, respectively (Figure 8c,d). In the bare group, the ECs exhibited mild movement of 68%, and the SMCs showed no significant difference. Furthermore, the sandwich coating, similarly to the control group, displayed migration rates of 82% for ECs and 38% for SMCs. The results were consistent with the cytocompatibility test, indicating that sandwich coating preserved the viability of ECs and inhibited the SMCs with the release of GA.

## 4. Discussion

One of the goals of surface modification of ZK60 is to enhance its corrosion resistance in simulated body fluid and hence improve its potential for use in fabricating bioresorbable coronary artery stents. Mg(OH)_2_ forms a passive layer on the magnesium surface through AHT [36]. This process protects the Mg surface from ion attack and prevents the accelerated degradation of ZK60 magnesium, which is supported by the polarization test in Figure A2. Based on the limitations in our lab, discussing the LA:GA ratio of PLGA and the coating techniques was beyond the scope of the study. A comprehensive degradation study on PLGA (50:50) and (85:15) revealed that complete degradation of the PLGA (50:50) occurred after 102 days, whereas only about 60% of the PLGA (85:15) degraded within the same period [37]. Therefore, we simply discuss the weight percentage of PLGA (based on the 85:15 ratio) for its long-term degradation process and chose the best group as the control group in this study. An oxide-passive layer on the Mg-disk led to a coarse surface until the second PLGA layer was deposited on it to fill the cracks (Figure 2), yet this passive layer prevented the initial hydrogen evolution from the ZK60 surface and prevented the acid degradation products of the PLGA layer from penetrating directly into the ZK60 substrate. This phenomenon explains why the NaOH passive layer is vital to make PLGA layer more completed and concrete on the coated surface. Although dip-coating might not be ideal for stent coating due to the complicated geometry, this technique can be simply conducted and qualifies for further bench testing, such as cytocompatibility and immersion test.

Many studies have investigated the use of phenolic molecules to regulate endothelialization based on their specific properties on SMCs and ECs [28,29]. These applications not only prevent the over-growth of SMCs from invading the EC wall but also assist in the formation of a uniform endothelial layer in the vascular system. Further, the goal of drug elution changed from anti-immunity to anti-proliferation since sirolimus (SR) replaced paclitaxel (PTX) as the main drug system for drug-eluting stent [38]. We believe our sandwiched GA concept can bring a novel direction to apply small-molecule on eluting stent instead of traditional SR or PTX. In this paper, the healing process can be nearly completed in 48 h, so a four-day viability test is consistent with the endothelial cell healing process (Figure 6 and Figure 8). Endothelial cells are unique in their growth pattern, preferring to grow in flat structures, so the extreme growth rate without an appropriate dynamic environment, a laminar blood flow, tends to cause apoptosis in limited living space [39,40]. Nevertheless, the opposite behavior between ECs and SMCs in this study suggests that the PLGA films promoted the proliferation of SMCs, while sandwich coating notably delayed them with similar behavior to that of the control group. Many recent clinical trials have revealed that polymer-based stents [18] do not mitigate late-stent restenosis, the SMC migration results may explain the deficiencies related to the use of polymers and offer a possible solution.

The anti-inflammatory effect is another concern related to cardiovascular stents. Phenolic molecules exhibit extraordinary, inherent anti-oxidative abilities, thus helping to scavenge reactive oxygen species and protect vascular tissue. In addition, free radical scavenging decreases oxidants and inhibits the atherogenesis initiated by oxidation of LDL. These antioxidant mechanisms prevent late myocardial infarction and in-stent restenosis after PCI. Although the drug delivery strategy is beyond the scope of our study, the interaction of free-radical capture and the GA delivery trend are what we are interested in. Recently, a free radical scavenging analysis completed by DPPH was conducted to evaluate the antioxidant capacity of the stent platform [34]. Phenolic molecules, such as ECGC and TA [28,29,30,31], have been discussed in similar studies, yet the GA application under PLGA eluting stents has not been sufficiently investigated. In our previous study, a phenolic-modified ceramic coating on ZK60 was proven to be efficient on osteo-like cell activity [12]. Due to its ability to capture free radicals and cell selectivity, GA can potentially be applied in the modified polymer coating. However, the GA content is hard to characterize in the host because of the circulation system, which is also correlated to the cell tolerance and apoptosis occurring in a static environment.

Under the limitations of a coating strategy for the complicated geometry of stents and the in vivo environment, only the surface modification and the interaction of materials and cells can be discussed within our scope. To optimize the scale of this research, other coating techniques, such as spray coating, and the in vivo environment will be discussed in the near future.

## 5. Conclusions

PLGA dip-coatings with gallic acid (GA) were prepared on a ZK60 surface in a sandwiched-layer structure. The a close-packed sandwiched layer showed enhanced corrosion resistance and a homogeneous film surface. An in vitro assay demonstrated that the sandwich coating behaved selectively with bioactivity on ECs, significant suppression of SMC over-proliferation, anti-hemolysis ability, and anti-oxidation effects compared to PLGA. This simple technique with a sandwiched-layer structure is a promising surface treatment for a commercialized stent platform for treating atherosclerosis and preventing late-stent restenosis.

## Figures and Tables

**Figure 1 materials-13-05538-f001:**
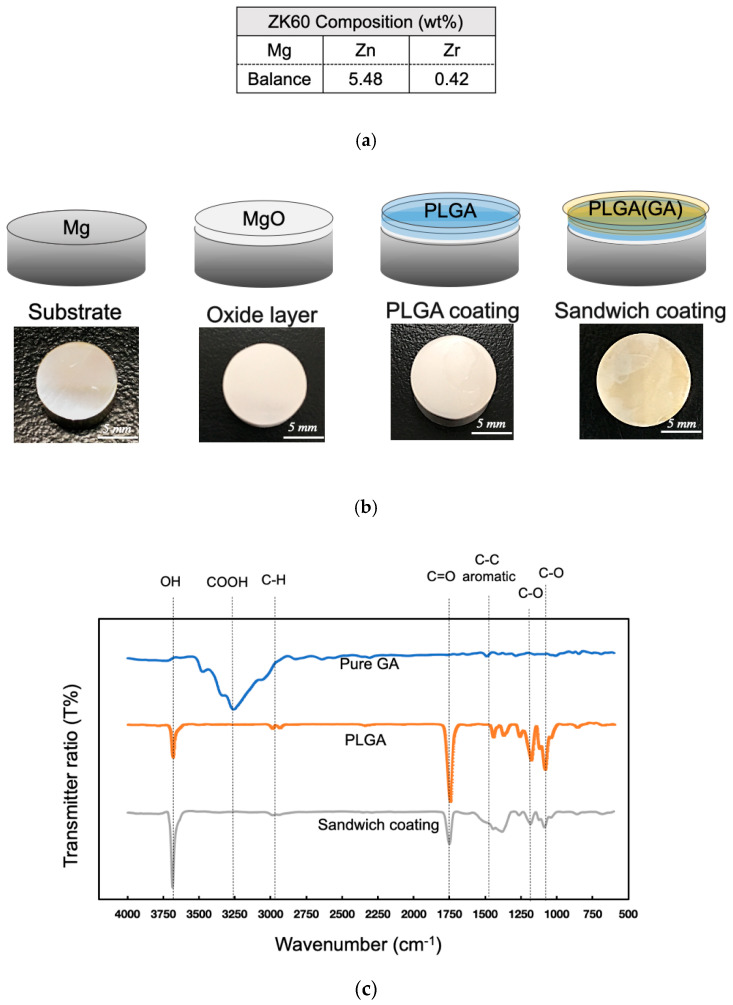
Composition of ZK60 (**a**) [34], the steps in the dip-coating procedure (**b**), and FT-IR spectra of GA, PLGA, and the sandwich coating (**c**).

**Figure 2 materials-13-05538-f002:**
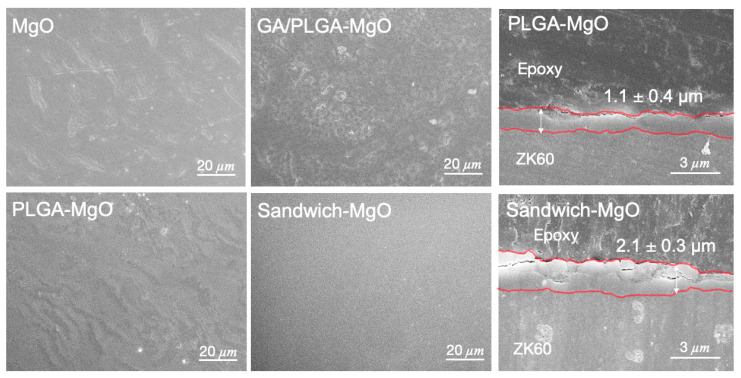
SEM images of the coating steps: MgO, PLGA-MgO, GA/PLGA-MgO, and Sandwich-MgO, the cross-section images of PLGA-MgO (1.1 ± 0.4 µm) and Sandwich-MgO (2.1 ± 0.3 µm).

**Figure 3 materials-13-05538-f003:**
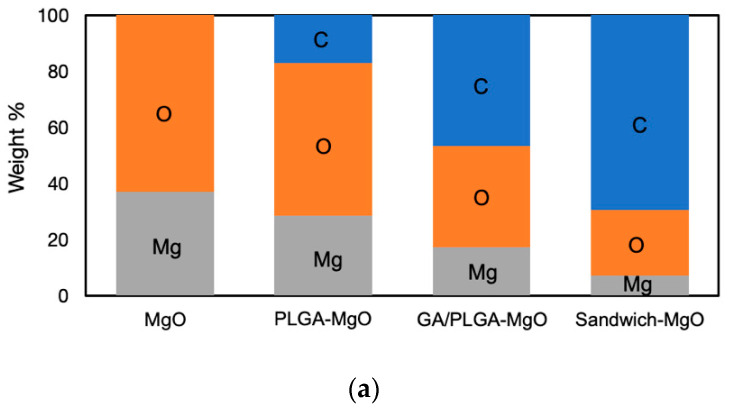
EDS composition analysis of (**a**) different layers and cross-sections of the (**b**) PLGA, and (**c**) sandwich coating films.

**Figure 4 materials-13-05538-f004:**
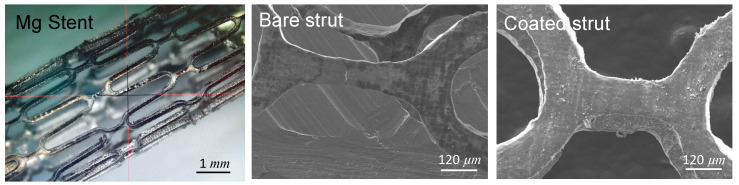
Mg-stent prototype, and the SEM images of the bare strut and sandwich-coated strut.

**Figure 5 materials-13-05538-f005:**
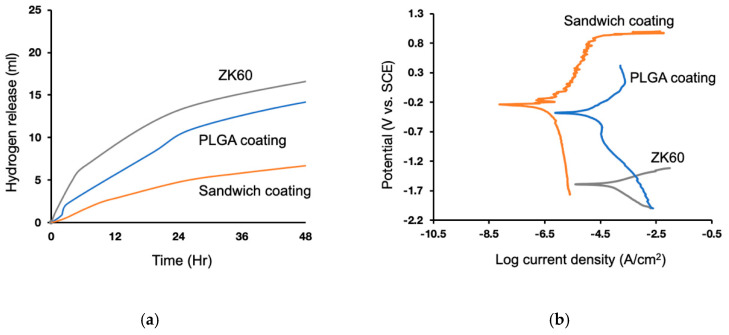
Corrosion resistance tests: (**a**) Hydrogen evolution curves of ZK60, PLGA coating and sandwich coating and (**b**) Potentiodynamic polarization curves of ZK60, PLGA coating, and sandwich coating in SBF solution.

**Figure 6 materials-13-05538-f006:**
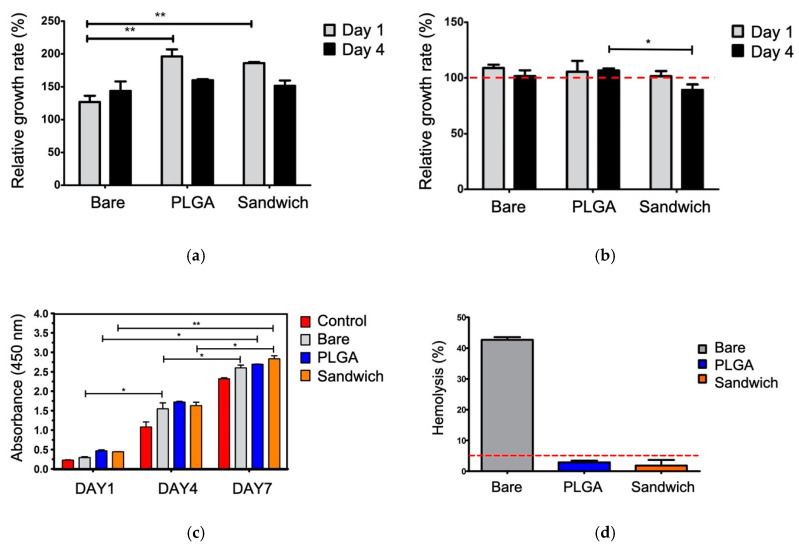
The cell viability evaluation of (**a**) ECs and (**b**) SMCs on day 1, day 4, (**c**) coatings versus EC proliferation (* *p* < 0.05 and ** *p* < 0.01, mean ± SD, N = 3) and (**d**) the hemolysis test.

**Figure 7 materials-13-05538-f007:**
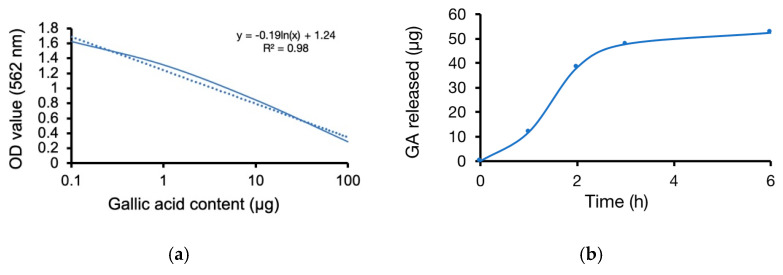
The anti-oxidation analysis of (**a**) the DPPH/GA efficiency curve, (**b**) the sandwich coating film release GA curve and (**c**) free radical scavenging activity (*** *p* < 0.001, mean ± SD, N = 6).

**Figure 8 materials-13-05538-f008:**
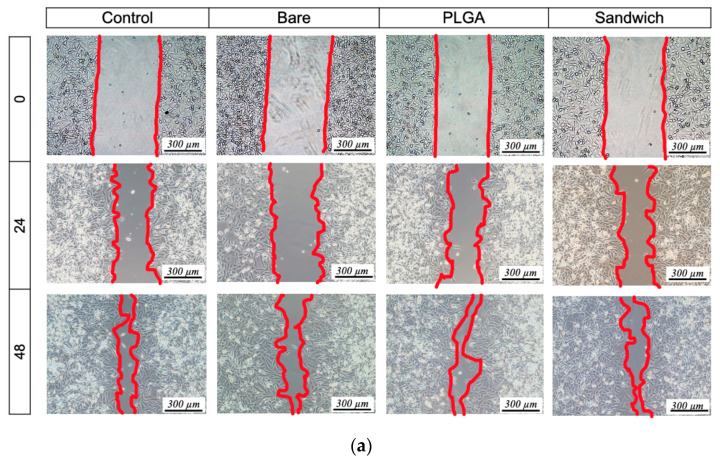
Cell migration of (**a**) EC, (**b**) SMC at 0, 24, and 48 h, and the fitting results for (**c**) EC and (**d**) SMC migration coverage (mean ± SEM, N = 3).

**Table 1 materials-13-05538-t001:** Fitting results for the potentiodynamic polarization curves related to Figure 5b. The inhibition efficiency was calculated as “η% = [1 − (Icorr. sample/Icorr. Bare ZK60)] × 100%”.

Specimen	Polarization Curves			
	Ecorr (V)	Log Icorr (µA/cm^2^)	Icorr (µA/cm^2^)	η (%)
Bare ZK60	−1.59	−4.69	20.51	0.00
PLGA coating	−0.40	−5.75	1.79	91.27
Sandwich coating	−0.24	−8.00	0.01	99.95

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
