# Peer review of "Characterization of a Sandwich PLGA-Gallic Acid-PLGA Coating on Mg Alloy ZK60 for Bioresorbable Coronary Artery Stents"

_materials, 2020, doi:10.3390/ma13235538_

Round 1
Reviewer 1 Report
First of all I would like to thank you for considering our journal to publish your manuscript. I have read your manuscript with great interest. The paper contains material which is worthy of publication and of potential interest to readers The manuscript entitled Anti-corrosion and Enhanced Endothelialization of a 3 Polyphenol-loaded PLGA Coating for Magnesium 4 Alloy Cardiovascular Stents presented by the authors is a very well-crafted document with numerous data supporting the discussion of the results.
I only find a small problem with the resolution of Figure 2, this can be significantly improved.
Author Response
Dear Reviewer,
Thank you for your review and advice. We sincerely appreciate it and have tried our best to answer your questions. Thank you.
Best Regards,
Li-Han Lin

Reviewer 2 Report
The submitted manuscript entitled “Anti-corrosion and Enhanced Endothelialization of a Polyphenol-loaded PLGA Coating for Magnesium Alloy Cardiovascular Stents” deals with an interesting topic of a coating for Mg alloy, which combines both corrosion protection with a biologically active function. The paper is well structured and clearly written and can extend the existing knowledge on this topic. Simultaneously, I think it matches the scope of the journal Materials. Therefore I recommend it for publication in its present form.
Author Response
Thank you for reviewing our article and your encouraging words. We sincerely appreciate your time and efforts.Reviewer 3 Report
This article describes the anti-corrosion and enhanced endothelialization properties of polyphenol-loaded PLGA coating prepared on ZK60 magnesium alloy. The reported experiments and results are seasonable and have some interest, and maybe published in Materials after major revisions.
It is worth publishing in Materials after some parts being improved.
Some units or collocations are not in the correct form:
Line 110: 1 mV-1;
Line 112: three-electrode electrode
Line 239: the unit of Log current density is incorrect
Line 76: There is no information about the processing of commercial ZK60 magnesium alloy (casting, wrought…).
Line 76: Is the listed chemical composition determined by norm or by your measurement?
Line 81: Why was used the cleaning medium in order of acetone->ethanol -> distilled water?
Line 82: There is no information about the polishing process.
Line 92-93: The form of drawing the PLGA thin film is listed in the manuscript without specifying more details. Please describe the coating process in more detail.
Line 94: “The samples were dipped into GA solution at a concentration of 1 wt.% and 10 ml acetone“.
Is the conjunction „and“ correct?
Line 105: There are missing the name of the electrochemical corrosion test. There is missing information about the time of exposure in SBF solution before the potentiodynamic polarisation was started and about the volume of SBF solution used for tests. It is not specified if the measurement of potentiodynamic polarisation tests started at open circuit potential. Did the hydrogen evolution tests and potentiodynamic polarization tests were performed simultaneously or separately? How many tests were released within one sample type?
Line 123: What is FBS DMEM?
Line 127: What are OD and CCK8?
Line 140: What is ELISA?
Line 144: What is DPPH
Line 161: What is AHT?
Line 201: Are the listed errors of the coating thickness measurement correct? Describe the method of coating thickness measurement.
Line 202: The description of the figure is defective. The description must be in order from base material to the coating top.
Line 206-207: There are missed measurement errors of EDX analysis. Additionally, the areas of the EDX analysis must be shown or described.
Line 206-207: What do you mean by the “level of coating”?
Line 217: The places of the coating with determined chemical composition measured by EDX (Figure 3. (a)) should be marked in Figure 3(b) and Figure 3(c).
Line 219-224: This part is not in a logical connection with other parts of the manuscript. The stent is not specified in the part of the Materials and Methods; what was the corrosive medium; what is the plasma film….
Line 239: Figure 5(b) the evaluation of potentiodynamic polarization curves seems to be incorrect. Could you send me the graph with lines of Taffel analysis?
Line 316; 318: Figure 8(a,b) the description of areas in Figures should be defined.
Author Response
Dear Reviewer,
Thank you for your review and advice. We sincerely appreciate it and have tried our best to answer your questions. Attached please find our revised manuscript, with the "response to reviewer's comments" placed at the end of the manuscript. Due to the file number limitation (only one file allowed) in the submission system, we are unable to attach the EDS reports and potentiodynamic tests analysis. Thank you.
Best Regards,
Li-Han Lin

Reviewer 4 Report
PLEASE SEE THE ATTACHED FILE, "SANDWICH-COATING-MG-ALLOY".

Author Response
Dear Reviewer,
Thank you for your review and advice. We sincerely appreciate it and have tried our best to answer your questions. Attached please find our revised manuscript, with the "response to reviewer's comments" placed at the end of the manuscript. Thank you.
Best Regards,
Li-Han Lin

Reviewer 5 Report
Their authors
My comments are given as an attachment.
The manuscript needs several improvements before further evaluation.
Regards.

Author Response

(The authors gave the same response as above.)

Round 2
Reviewer 5 Report
Dear authors
The manuscript was corrected in accordance with my suggestions.
There remain some minor comments/suggestions (see attachment).
After their consideration, the manuscript can be accepted.
Nice work.
Regards.

Author Response
Dear Reviewer,
Thank you for your review and advice. We sincerely appreciate it and have tried our best to answer your questions. Attached please find our revised manuscript.
Thank you.
Best Regards,
Li-Han Lin
